# Assessment the Impacts of Sea-Level Changes on Mangroves of Ceará-Mirim Estuary, Northeastern Brazil, during the Holocene and Anthropocene

**DOI:** 10.3390/plants12081721

**Published:** 2023-04-20

**Authors:** Sérgio. P. D. Q. Nunes, Marlon C. França, Marcelo C. L. Cohen, Luiz C. R. Pessenda, Erika S. F. Rodrigues, Evandro A. S. Magalhães, Fernando A. B. Silva

**Affiliations:** 1Laboratory of Coastal Dynamics, Graduate Program of Geology and Geochemistry, Federal University of Pará, Belém 66075-110, PA, Brazil; 2Federal Institute of Espírito Santo, Piúma 29285-000, ES, Brazil; 3Department of Oceanography and Coastal Sciences, College of the Coast and Environment, Louisiana State University, Baton Rouge, LA 70803, USA; 4CENA/14C Laboratory, University of São Paulo, Av. Centenário 303, Piracicaba 13400-000, SP, Brazil

**Keywords:** drone, Holocene, palynology, relative sea-level, remote sensing, sedimentary facies

## Abstract

Predictions of the effects of modern Relative Sea-Level (RSL) rise on mangroves should be based on decadal-millennial mangrove dynamics and the particularities of each depositional environment under past RSL changes. This work identified inland and seaward mangrove migrations along the Ceará-Mirim estuary (Rio Grande do Norte, northeastern Brazil) during the mid–late Holocene and Anthropocene based on sedimentary features, palynological, and geochemical (δ^13^C, δ^15^N, C/N) data integrated with spatial-temporal analysis based on satellite images. The data indicated three phases for the mangrove development: (1°) mangrove expansion on tidal flats with estuarine organic matter between >4420 and ~2870 cal yrs BP, under the influence of the mid-Holocene sea-level highstand; (2°) mangrove contraction with an increased contribution of C_3_ terrestrial plants between ~2870 and ~84 cal yrs BP due to an RSL fall, and (3°) mangrove expansion onto the highest tidal flats since ~84 cal yr BP due to a relative sea-level rise. However, significant mangrove areas were converted to fish farming before 1984 CE. Spatial-temporal analysis also indicated a mangrove expansion since 1984 CE due to mangrove recolonization of shrimp farming areas previously deforested for pisciculture. This work mainly evidenced a trend of mangrove expansion due to RSL rise preceding the effects of anthropogenic emissions of CO_2_ in the atmosphere and the resilience of these forests in the face of anthropogenic interventions.

## 1. Introduction

Global mangrove distribution, which occurs along tropical and subtropical latitudes, has changed in the Holocene and Anthropocene [1,2,3]. These forests are influenced by a complex interaction mainly involving changes in sea-level, temperature, and fluvial discharge [4,5,6,7,8]. Along the northern Brazilian coast, fluvial discharge and sea-level changes have been the major forces driving the mangrove dynamics in the late Pleistocene and Holocene [5,6,9,10,11,12,13,14,15]. However, Relative Sea-Level (RSL) changes, with a highstand between 1 and 5 m above the modern sea-level (amsl) at ~5500 cal yr BP [5,16,17,18,19,20], have been the primary mechanism driving mangrove spatial dynamics between the northeastern and southern Brazilian coast during the Holocene [5,10,14,21]. The RSL on the Rio Grande do Norte coast in northeast Brazil (Figure 1) reached the current level at ~7000 cal yr BP [22] with the highstand (~1.3 m) at ~5900 cal yr BP [20]. According to [16], an RSL fall trend occurred between the southern and northeastern coasts of Brazil during the late Holocene.

Besides natural forces, the anthropogenic conversion of mangrove forests to land meant for aquaculture/agriculture has caused global mangrove loss [23]. The mangroves and herbaceous flats at the mouth of the Ceará-Mirim River in northeastern Brazil, despite being impacted by aquaculture, exhibit typical mangrove zonation across topographical gradients [22], where the mangrove/herbaceous flat transition is highly susceptible to changes in the tidal flooding regime [24]. The Rio Grande do Norte coast is influenced by intense winds, waves, tidal currents, fluvial discharge, and sediment input [25]. According to [22], mangroves in the Ceará-Mirim River, north of the city of Natal, presented an ecological history influenced mainly by autogenic mechanisms related to tidal channel dynamics, instead of allogenic processes related to climate changes since ~6920 cal yr BP. We hypothesize that the studied mangroves may have responded to changes in the relative sea-level and river discharge since the mid Holocene. The effects of sea-level and fluvial discharge on the mangroves of Natal have never been recorded and addressed in the literature. However, this question is vital for understanding and predicting future mangrove dynamics under natural and anthropogenic influences. Therefore, how did the morphology of an estuarine mangrove, under natural (fluvial discharge and sea-level) and anthropogenic (pisciculture) influence, respond to sea-level changes in the Holocene and Anthropocene? To answer that question, we chose a methodology that combines pollen, isotopes (δ^13^C, δ^15^N), and elemental (C\N) analyses along topographically referenced cores sampled from the highest and lowest tidal flats of the Ceará-Mirim estuary with historical low (Landsat) and high (QuickBird and Drone) spatial resolution images and aerophotogrammetric data acquired by drones. Drone data allowed us to identify the coastal vegetation structure associated with the topographical gradients of the coastal plain. The application and analysis of drone images (3 cm resolution) permitted the identification of small patches of mangroves and their changes in much greater detail [14,26,27,28,29,30]. This enabled a more comprehensive characterization of the coastal vegetation zonation in a planialtimetric perspective than using in situ surveys alone.

## 2. Results

### 2.1. Vegetation and Geomorphological Units

The study area presents a fluvial valley under the influence of the Ceará-Mirim River (Figure 1), which is born in the Coastal Plateau and flows on the Coastal Plain along a topographic gradient of 90 m. Some channels were rectified due to shrimp farming activities, which occupy ~2.8 km^2^, causing a significant impact on the wetlands vegetation (Figure 1). Mangroves (~5 km^2^), with *Rhizophora mangle* and *Avicennia germinans*, occur along ~5 km of the estuarine channel on tidal flats between 1 and 2 m above mean sea-level (amsl). *Rhizophora* trees occur mainly on lower tidal flats (~1 m amsl), showing the tallest trees (~15 m tall), while the smallest *Rhizophora* (~2 m tall) inhabit middle elevations (~1.7 m amsl) (Figure 2a). *Avicennia* trees occur mainly on the highest tidal flats (1.7–~2 m amsl) and present the highest trees (~7 m tall) on lower surfaces (~1.7 m amsl), causing a transition with the lowest *Rhizophora*. The tallest *Avicennia* trees (~10 m) are spread among the tallest *Rhizophora* on the lowest flats (~1 m amsl). Some *Laguncularia* (~5 m tall) are spread in smaller numbers, mainly among *Avicennia* trees on the highest flats. Herbaceous vegetation (4.6 km^2^), represented by *Poaceae*, *Cyperaceae*, *Borreria* and some *Arecaceae*, occurs on the highest tidal flats (2–3 m amsl). *Arecaceae* trees inhabit seasonally and permanently inundated flats (~7 km^2^) by freshwater, while the coastal plateau presents wooded steppe savannah mainly characterized by *Cyperaceae* and *Poaecea*, with few shrubs primarily characterized by *Anacardiaceae* and *Malpighiaceae* (Figure 1).

### 2.2. Multitemporal Analysis

Satellite images indicated a mangrove expansion trend between 1984 and 2018, with 315,800 ha in 1984 increasing to 443,400 ha (2000), 404,800 ha (2005), 483,200 ha (2011), 435,100 ha (2016), and 590,200 ha (2018). Mangroves expanded mainly onto herbaceous flats and abandoned shrimp farms (Figure 3). The mangrove expansion was not progressive and constant mainly on higher limits of tidal flats (~1.7–2 m amsl), where *Avicennia* trees led this mangrove migration onto the highest tidal flats. The transition zones show advances and retreats of the mangrove/herbaceous vegetation, where there is the coexistence of Poaceae, Cyperaceae, and *Avicennia* shrubs. It is noteworthy that isolated and elevated sandy tidal flats (~2 m amsl), presenting a circular morphology with herbaceous vegetation and without human interference, also show intermittent advances and retreats of *Avicennia* shrub limits, but with an expansion trend between 1984 and 2018. By contrast, a progressive mangrove expansion has occurred mainly onto lower limits of tidal flats (~1–~1.7 m amsl) with abandoned shrimp farms. These lower surfaces have been consistently invaded, mainly by *Rhizophora*.

### 2.3. Radiocarbon Dates

The C−14 dating for the NAT−3 and NAT−5 cores are exhibited in Table 1. The RBacon model for the C−14 ages allowed us to infer two ages in the core NAT−5: ~2170 cal yr BP at 55 cm, and ~84 cal yr BP at 5 cm (Appendix A, Figure 4).

### 2.4. Facies Description

The radiography of these cores allowed us to identify internal sedimentary structures. As recorded in Figure 5 and Figure 6, it would not be possible to evidence such structures without radiography. The heterolytic beddings show continuous vertical sediment accretion with some bioturbation, but no significant sedimentary reworking that would weaken the paleoenvironmental reconstruction. Thus, we assume these sediments, pollen content, and sedimentary organic matter originated from establishing the depositional environment with terrestrial and aquatic vegetation. Regarding the palynology, we can consider two components—pollen from ‘‘local’’ vegetation and background pollen from ‘‘regional’’ vegetation [31,32,33]. The transition between the local and regional is gradual and depends on each pollen taxon [34,35]. The isotopic and elemental characteristics of sedimentary organic matter along the studied cores depend on the in situ vegetation and input of fluvial and marine organic matter [36,37,38]. Then, the sedimentary sequences presented massive sand (Sm), massive mud (Mm), and wavy (Hw) heterolytic beddings. The sedimentary features; pollen records; and δ^13^C, δ^15^N, TOC, TN, and C:N data allowed us to propose three facies associations to determine sedimentary environments [39]: (A)—Tidal channel/sandbar; (B)—herbs/mangrove mixed tidal flat; (C)—Herbaceous plain (Table 2). Cluster analysis, based on pollen percentages, contributed to defining facies associations along the core NAT−5, while this analysis did not show significant pollen percentage changes that justified more than one facies association along the core NAT−3.

### 2.5. Facies Association A (Tidal Channel/Sandbar)

This facies association presents massive sand (Ms) (50–80% sand, 2–45% silt, 1–5% clay). This facies association occurs at the basis of the NAT−5 in intervals of 150–130 cm depth (Figure 5). Herb pollen grains (~40%), mainly represented by *Cyperaceae* (0–33%), *Poaceae* (0–23%), *Euphorbiaceae* (0–20%), *Typhacea* (0–4%), and *Cucurbitaceae* (0–4%), predominate in this facies association. Trees and shrubs (35 to 71%) are mainly represented by *Fabaceae* (0–40%), *Moraceae/Urticaceae* (0–25%), *Rubiaceae* (0–38%), *Myrtaceae* (0–23%), *Bignoniaceae* (0–5%), *Melastomataceae* (0–5%), *Meliaceae* (0–4%), *Myristicaceae* (0–4%), *Malpighiniaceae* (0–3%), and palms, formed by *Arecaceae* (0–6%). Mangrove pollen (15 to 47%) are represented by *Rhizophora* (0–35%), *Laguncularia* (0–40%), and *Avicennia* (0–3).

The δ^13^C and δ^15^N values vary between −29.7‰ and −26.8‰ (mean 28.2‰) and from −0.84 to 1.6‰ (mean 1.22‰), respectively. TOC and TN values are between 3.74 and 17.58% (average 10.66%) and 0.12 and 0.54% (average 0.33%), respectively. The C:N values occur between 21.2 and 45 (Figure 4).

### 2.6. Facies Association B (Herbs/Mangrove Mixed Tidal Flat)

These deposits are characterized mainly by wavy (Hw) heterolytic bedding (Hl) and by massive mud (Mm) (20–45% sand, 48–60% silt, 2–20% clay) with bioturbation, represented by plant fragments (Figure 5 and Figure 6). Roots arranged horizontally and vertically partially obliterate the sedimentary structures, impairing the identification of heterolytic beddings on radiographs. It occurs between 130–55 cm (NAT−5, ~4420–~2170 cal yr BP) and 5–0 cm depth (NAT–5, ~84 cal yr BP—today), and between 55 and 40 cm depth (NAT−3). Pollen analysis indicated mangrove presence (13 and 30%), mainly represented by *Rhizophora* (8–30%). Herbs (45–82%) are mainly characterized by *Cyperaceae* (40–68%), *Poaceae* (0–7%), and *Typhacea* (0–3%). Trees and shrubs (17–38%) are characterized by *Fabaceae* (0–34%), *Anacardiaceae* (0–22%), *Euphorbiaceae* (0–20%), *Bignoniaceae* (0–8%), *Moraceae/Urticaceae* (0–20%), *Rubiaceae* (0–15%), *Myrtaceae* (0–8%), *Melastomataceae* (0–6%), *Meliaceae* (0–5%), *Theaceae* (0–7%), and *Ilex* (0–4%). *Arecaceae* pollen occurs with 1–5%. The δ^13^C and δ^15^N values are between −29.2 and −28.7‰ (mean 28.95‰) and −0.91 and 2.91‰ (mean 1‰), respectively. TOC and TN present values between 3.44 and 17.58% (mean 10.51%) and 0.06 and 0.43% (mean 0.24%), respectively. The C:N values are between 27.6 and 34.14 (Figure 4 and Figure 7).

### 2.7. Facies Association C (Herbaceous Tidal Flat)

This facies association consists of massive mud (Mm) and wavy (Hw) heterolytic bedding (5–15% sand, 80–90% silt, 5–15% clay) and the presence of fragments of plants and roots (Figure 5). It occurs between 55 and 5 cm depth (NAT−5, ~ 2170–~84 cal yr BP). This facies association reveals herb pollen (67–87%) represented by *Cyperaceae* (26–50%), *Poaceae* (0–27%), *Euphorbiaceae* (0–20%), *Typhacea* (0–25%), *Mimosa* (0–6%), and *Asteraceae* (0–5%). Trees and shrubs (25–40%) are represented by *Fabaceae* (0–22%), *Anacardiaceae* (0–24%), *Moraceae/Urticaceae* (0–10%), *Rubiaceae* (0–5%), and *Myrtaceae* (0–10%). Mangroves, represented by *Rhizophora* (0–1%), and palms, characterized by *Arecaceae* (0–3%), were also identified. The δ^13^C and δ^15^N values are ~−28.9‰ and ~0.58‰, respectively. TOC and TN present percentages between 4.25 and 20.4% (mean 12.32%) and 0.43 to 0.83% (mean 0.63%), respectively. The C:N values are between 24.5 and 28.67 (Figure 4).

## 3. Discussion

The interpretations of this study are based on robust multi-proxies. Environmental proxy indicators have the potential to provide evidence for large-scale climatic, sea-level, and depositional system changes. However, the interpretation of a proxy record is complicated by the presence of “noise” in which environmental information is immersed and there is a variety of possible distortions of the underlying environmental information [40,41]; then, a high-resolution of a set of proxies should be used to provide detailed information on the environment back in time [42] We believe that paleoenvironmental reconstructions based on facies association (e.g., pollen, isotopes, and sedimentary features) along stratigraphic profiles (NAT−3 and NAT−5), integrated to the characterization of current vegetation and geomorphological units of the study area, have the potential to attenuate such “noise” in the signals of the paleoenvironmental indicators, making the interpretations more reliable.

### 3.1. Depositional Phases

Multi-proxy data suggest four phases of mangrove development: The first stage is represented by massive sand (Sm) with a gradual transition to wavy heterolithic beddings (Hw), suggesting that flow energy decreased [43,44]. These facies may occur in various depositional settings [45,46], but considering the studied sedimentary environment with tidal flats, tidal/estuarine channels, and sandbars, under the influence of a meso-tide (2–4 m, [47]), it would be reasonable to propose that transition was caused by typical tidal channel dynamics building up an upward-fining succession, with thick sand deposition succession at the base, including subtidal channel-filling, topped by intertidal muddy deposits. Alternatively, subtidal sandbars may have been exposed to a decreased flow energy and gradually began to accumulate wavy heterolithic bedding. Pollen analysis indicated herbaceous vegetation with *Rhizophora, Avicennia*, and *Laguncularia,* typical elements that occur along the margin of a tidal channel (facies association A). Isotopic data revealed that organic matter was sourced from terrestrial C3 plants and estuarine organic matter >4420 cal yr BP. The second phase (~4420–~2170 cal yr BP) in the NAT−5 site (2.5 m above mean sea-level) is characterized by the establishment of the mixed tidal flat dominated by herbaceous vegetation and mangrove expansion, represented by *Rhizophora*. This phase is represented by estuarine organic matter (Figure 8a). The third phase is marked by a mangrove contraction and an increase in the contribution of terrestrial organic matter between ~ 2170 and 84 cal yr BP in the highest tidal flats (~2.5 m above mean sea-level, amsl) and a mangrove expansion in the lowest tidal flats (~1 m amsl) during the last 600 cal yr BP (Figure 2a, Figure 4, Figure 7, Figure 8 and Figure 9). The fourth phase is characterized by mangrove expansion in the highest tidal flats (~2.5 m amsl) during the last 84 cal yr BP (Figure 4 and Figure 9).

### 3.2. Mid-Holocene High Sea-Level Stand

RSL changes are related to land-based reference [50,51], mainly controlled by eustatic sea-level changes, glacio-isostatic adjustment, gravitational attraction, and tectonics tectonic [52,53,54,55]. The upper (~2.5 m amsl) and lower (~1.5 m amsl) tidal flats with mangroves were used as a reference to propose RSL changes in the studied area. Isotopic (δ^13^C and δ^15^N) and C/N analyses combined with mangrove pollen grains along stratigraphic profiles have been used to record trends of RSL changes [5,6,9,12,13,15,56,57,58], because these isotopes and elemental analyses indicate the source of sedimentary organic matter, mainly between end-members: terrestrial- and aquatic-derived organic matter [38,59,60,61,62]. Then, the δ^13^C, δ^15^N, and C/N changes may indicate an RSL trend [58]. Additionally, mangrove forests occur along a topographical gradient within the modern tidal range [9,30], accumulating mangrove pollen [34], which allows researchers to record sediments accumulated under mangrove vegetation, defining a proxy-record of RSL changes [5,9,63].

In this context, RSL rose on the northeast coast of Brazil during the early and mid Holocene [5,16,64] with a high sea-level stand at about 5500 cal yr BP (1–4 m amsl), favoring the mangrove establishment on the highest tidal flats and expansion upriver, as evidenced in core NAT–5 (>4420–2870 cal yr BP) (Figure 2a and Figure 4), where today there is a transition zone between mangrove and palms/herbaceous vegetation. The C/N and δ^15^N values oscillated between 45 and 18 and 0 and 3, respectively, suggesting a blend of aquatic and terrestrial organic matter. A paleoecological study revealed mangrove establishment in the study area at ~6920 cal yr BP [22] after a rapid RSL rise between 8300 and 7000 cal yr BP in Rio Grande do Norte ([20]. According to [64], a relatively rapid sea-level rise occurred in Rio Grande do Norte between ∼7100–5800 cal yr BP and about 5000 cal yr BP, reaching 2.5–4.0 m above present sea-level. Geochemical and sedimentological analyses along cores sampled from the Parnaiba Delta (780 km northwest of the study area) indicated that the oldest paleo-mangrove soil was at 4853 to 4228 cal yr BP [65], while multi-proxy data revealed that mangroves occupied the highest tidal flats of the Bragança Peninsula (1400 km northwest of the study area) between ~6250 and ~5850 cal yr BP (with a highstand of 0.6 ± 0.1 m above msl at ~5000 cal yr BP) [9]. RSL reached a highstand along the Suriname and Guyana coasts of 1.0 ± 1.1 m between 5.3 and 5.2 ka [66]. The highest RSL in the Holocene (1–5 m above the modern sea-level (amsl)) occurred at ~5500 cal yr BP along the southeastern and part of the northeastern Brazilian coast [5,16,17,18,19], corroborating with the mangrove establishment on the north–northeast Brazilian coast, after the post-glacial sea-level rise, in the mid Holocene.

### 3.3. Sea-Level Fall during the Late Holocene

Studies have indicated a gradual RSL fall in southeastern, northeastern, and northern Brazil since the mid-Holocene high sea-level stand [5,9,16,18,67] or oscillations that may have occurred [19,20,64]. The effects of the RSL fall on coastal vegetation and sedimentary organic matter in the study area, with a reduction in marine influence and seaward mangrove migration, occurred only after ~2870 cal yr BP. The lower mid-Holocene high sea-level stand (~1 m amsl) along the north and northeast Brazilian coast [9,20], when compared with the southeastern and southern Brazilian coast (3–5 m amsl, [16,18,67]), may have caused the delay in recording this RSL fall in the studied cores. In addition, the climatic component through fluvial discharge may have interfered with the effects of marine influence upstream (see Section 3.4. Climatic effects).

The transition from mid-Holocene high sea-level stand to RSL fall triggered feedbacks associated with tidal range, vertical sediment accretion, and tidal water salinity along the estuarine channel [9]. Mangrove areas may contract, expand, or be re-established at a lower surface under these new environmental conditions [5,9,68,69,70]. The reduced marine influence allowed an expansion of herbs, trees, and shrubs on higher flats and mangroves on lower surfaces (Figure 9), as evidenced in cores NAT–3 (Figure 7) and NAT–5 (Figure 4). The mangrove pollen percentage decreased from 35% to 0%, and the herb pollen increased from 50% to 80% after ~2870 cal yr BP in core NAT–5. The C/N values increased from 15 to 42, and δ^15^N values decreased from 3 to 0, indicating a decreased trend in the aquatic influence. The core NAT–3, sampled from a tidal flat lower (~1.5 m amsl) than core NAT–5 (~2.5 m amsl), presented an increased trend of mangrove pollen percentage from 15 to 35% during the last ~600 cal yrs BP, probably due to an RSL fall, which favored mangrove expansion on lower surfaces.

After the mid-Holocene sea-level highstand (∼5000 cal yr BP), RSL fell immediately and eventually rose again about 2100–1100 cal yr BP, resulting in a second coastal retreat in the Holocene [64]. This curve also indicates that a lower than actual RSL could have occurred in the Rio Grande do Norte. The studied cores did not indicate RSL oscillations between 2100 and 1100 cal yr BP. However, our data also do not allow refuting the possibility of an RSL oscillation during the late Holocene, because the sedimentary deposits, where the cores were sampled, may not have preserved pollen and geochemical evidence of this rapid and temporally short RSL rise. Following the topographic and estuarine gradient, new cores along the estuary could contribute to studies about late-Holocene RSL oscillations along the Brazilian coast [16]. In addition, studies have shown that climatic changes associated with RSL fluctuations have significantly impacted mangrove dynamics along the north–northeastern Brazilian coast during the Holocene [5,6,9].

### 3.4. Climatic Effects

Mangrove retraction on the highest tidal flats of the study area may not be exclusively influenced by an RSL fall. An increase in rainfall regime on mangroves and over the Ceará-Mirim River basin may cause a decrease in the estuarine salinity indicators and mainly affect the values of δ^13^C, as recorded along the northern and northeastern Brazilian coast throughout the late Holocene [5,6]. This process may replace mangroves with freshwater vegetation. Considering a sea-level fall and a rise of fluvial discharge working together in the late Holocene, these factors would be in phase promoting a mangrove migration from highest to lowest tidal flats. Therefore, the climate is an influence that must be considered in assessing the Holocene mangrove dynamics in the Ceará-Mirim River. Considering the climatic hypothesis, the aquatic organic matter in NAT−5 should have remained stable during the last ~4420 cal yr BP. However, this core presented an increase in C/N (from 15 to 42) and a decrease in δ^15^N (from 3 to 0) values, suggesting a decreased trend in the aquatic influence, then a significant impact of a relative sea-level fall in the source of sedimentary organic matter in the NAT−5 site (the highest tidal flat of the study area) during the last ~2170 cal yr BP. Furthermore, climate trends in northeastern Brazil during the mid–late Holocene are still open to debate. In the Maranhão littoral, ~800 km distant from the study area, paleoecological studies revealed a drier climate between ~10,000 and ~4000 cal yr BP [71], and a wet period from 4000 cal yr BP to the present [71,72]. Analogous climatic circumstances have been evidenced on the Brazilian coast during the Holocene [5,6,73,74,75,76]. A wet period occurred along the Brazilian (east) coast during the last millennium [77,78,79]. However, some researchers highlight that rainfall regimes in northeastern and southeastern Brazil are anti-phased during the Holocene [80]. The paleo-precipitation history of the Rio Grande do Norte speleothem indicates an “anti-phased” relationship to the rest of tropical South America [81,82,83]. Therefore, the northeastern Brazilian climate could be anti-phased compared with tropical South America, but a lack of paleo-data in the Rio Grande do Norte does not allow a more robust assessment to identify which climatic pattern prevailed in the study area.

### 3.5. Recent Sea-Level Rise

The core NAT−5 indicated a mangrove reoccupation trend on the highest flats of the study area since ~84 cal yr BP (Figure 4 and Figure 5). Currently, the highest tidal flats (>2 m amsl) are occupied by herbaceous vegetation (Figure 2a,b). In comparison, the mangroves dominated by the lowest *Avicennia* trees (<5 m tall) occur on tidal flats between 1.7 and 2 m amsl. The lowest tidal flats (~1 m amsl) are dominated by *Rhizophora* (Figure 2a,c). This zoning is caused by a porewater salinity gradient, where more frequently flooded zones have a lower porewater salinity that approaches tidal water salinity (20–30‰, [84]) and are occupied by mangrove trees with low tolerance to high salinities, such as *Rhizophora*. It contrasts with the higher zones that present a lower tidal inundation frequency and greater exposure to evaporation, resulting in higher porewater salinity, and are dominated by herbs with high tolerance to hypersaline zones, followed by *Avicennia* mangrove trees with greater tolerance to high porewater salinities than *Rhizophora* [9,14,24]. This interaction between mangrove structure, porewater salinity, and tidal flat elevation is important for understanding and interpreting pollen profiles. According to pollen accumulation rates obtained from northern Brazil, *Rhizophora* trees, which are pollinated by wind and insects, release more pollen than *Avicennia* and *Laguncularia,* which are pollinated by insects [34]. Therefore, even pollen profiles from tidal flats with a dominance of *Avicennia* and *Laguncularia* present higher *Rhizophora* pollen percentages than *Avicennia* and *Laguncularia*. The increase in the pollen percentage of *Rhizophora* in NAT−5 during the last 84 cal yr BP does not indicate the presence of these trees immediately above the sampling site, but rather the increase in proximity of *Rhizophora* mangroves to higher tidal flats, where this core was sampled. In addition, even the pollen profile shows a lower *Avicennia* pollen percentage than that of *Rhizophora* along the upper 5 cm (Figure 5); *Avicennia* trees are closer to the sampling site NAT−5 than *Rhizophora* trees, as identified in the modern mangrove trees distribution in the study area (Figure 2a). Therefore, the increase in *Rhizophora* pollen percentage suggests an incursion of the lower *Rhizophora* mangrove zone into higher tidal flats during the last 84 cal yr BP.

In this context, an RSL rise caused an increase in the tidal inundation frequency as the upper limit of the tides reached progressively higher surfaces. This process caused salinization of the higher flats, but with a gradual decrease in porewater salinity along the topographic gradient as the time available for evaporation on tidal flats decreased [5,6,9,24,30,84]. This process caused a decrease in porewater salinity on higher tidal flats since 84 cal yr BP, contributing to a decrease in the saline stress in the *Avicennia* and *Rhizophora* trees, and driving the mangrove zones to higher tidal flats. The landward mangrove migration onto higher tidal flats may be considered a global process, suggesting a common driving force likely related to sea-level rise driven by global warming [9,14,24,30,85,86,87,88,89]. However, the trend of relative sea-level rise identified in the study area precedes the effects of anthropogenic emissions of CO_2_ in the atmosphere. The relative sea-level rise in our study area could be related to the end of the Little Ice Age (LIA) that lasted roughly from 1300 until 1850 CE [90,91,92]. The impacts of this event have been identified in South America [12,14,93,94,95].

This trend of relative sea-level rise can be recorded even during the last decades [14,24,30], because spatial-temporal analysis of the study area revealed a mangrove migration onto tidal flats since 1984, establishing an increase in mangrove area from 315 ha in 1984 to 443 ha in 2000 and from 483 ha in 2011 to 590 ha in 2018 (Figure 3). However, mangroves in the study area have been largely deforested for pisciculture. The 1984 satellite image shows extensive shrimp farming pools in mangrove areas (Figure 1). Shrimp farming areas have been abandoned in recent decades, and mangroves are gradually recolonizing these tidal flats. Therefore, particularly in the study area, the recent sea-level rise may be contributing to the mangrove expansion, and despite the intense mangrove deforestation process before 1984 CE, these forests have shown great resilience to these anthropic interventions.

## 4. Materials and Methods

This study integrated planialtimetric and stratigraphic data, following a methodology flow chart divided into three phases (Figure 10): (1) spatial-temporal analysis based on satellite images; (2) choice of areas for a topographic survey and vegetation structure studies (stature and genera of mangrove trees) based on photogrammetry of drone images with field validation; (3) choice of the core sampling sites to pollen, isotope, sedimentary features, and C-14 analyses. This process allowed the integration and interpretation of surface and subsurface data to reconstruct the Holocene mangrove history.

### 4.1. Study Area

The researched area, situated in the mouth of the Ceará-Mirim River in northeastern Brazil (Figure 1), is characterized by a Precambrian crystalline basement, Cretaceous sedimentary rocks of the sedimentary basins, and by Miocene to Pliocene rocks of the Barreiras Formation [96]. Sediments accumulated during the Quaternary cover these basement rocks [97]. The Ceará-Mirim River basin is 2635.7 km^2^ and flows in a west–east direction, with a length of 120 km crossing the coastal plateau/plain. It is influenced by semidiurnal saline and dynamic mesotides with a mean spring and neap tidal range of ~2.30 m and 0.85 m, respectively. Near the mouth, it presents a transition zone under marine and fluvial influence, dominated by sand–clay and silty sediments [98]. The coastal plain is characterized by brackish water vegetation represented by mangroves on muddy tidal flats. Herbaceous vegetation occurs on the highest sector of the tidal flats. Upstream, sandy silt and muddy sediments spread over the fluvial plain, mainly dominated by “várzea” vegetation (swampland seasonally and permanently inundated by freshwater). The limit between várzea and mangrove vegetation depends on the estuarine salinity gradients [22,84]. Mangroves are characterized by *Rhizophora mangle*, *Avicennia schaueriana*, *Avicennia germinans*, and *Laguncularia racemosa.* Herbaceous vegetation is represented by typical species of the brackish environment, with halophytes that tolerate a high degree of salinity, mainly characterized by *Sesuvium portulacastrum* and *lresine vermicularis* [99]. Some topographically higher areas and hypersaline zones are colonized especially by shrub and herbaceous species, such as *Sesuvium portulacastrum, Conocarpus erectus*, *Laguncularia racemosa*, *Avicennia germinans,* and *Spartina alterniflora* [99]. The transition from mangrove to herbaceous flat that is periodically flooded is characterized by *Borreria* [100]. Floodplains upstream are occupied by pioneer formations, mainly represented by *Fabaceae*, *Rubiaceae*, *Arecaceae*, *Poaceae*, and *Cyperaceae* families [100,101]. Herbaceous substrate with few shrubs characterizes the wooded steppe savanna [100]—Figure 1. The climate of northeast Natal is a hot and humid tropical, with average monthly temperatures of ~26 °C. Average rainfall is about 129 cm per year.

### 4.2. Remote Sensing

Landsat 5 and 8 satellites (resolution of 30 m), with three bands (green, red, and near-infrared), were acquired from INPE (National Institute of Space Research, Brazil), Global Land Cover Facility Project Website—(GLCF), and US Geological Survey (USGS). These images (1984–2018) were used to identify vegetation and geomorphological units by an unsupervised classification with the PCI software. Ground control points using a GPS (Global Navigation Positioning System) provided a reliable indicator to subsidize the classification obtained by satellite images. Kappa index, with a sampling of 300 pixels for the classification, in random mode, followed the methodology of [102] to validate the classification. The confusion matrices indicated a percentage of pixels that were mapped correctly. However, objects smaller than the pixel size (900 m^2^, 0.09 ha) are indistinguishable. Therefore, the mapping of mangrove/herbaceous vegetation and mangrove/abandoned shrimp farms transition will be limited by the uncertainties caused by the pixels with mixed coverages (e.g., mangrove, herbs, soil, and water). The spectral response of these coverages depends on the average reflectance involving the different targets in the pixel [103].

### 4.3. Topographic and Vegetation Height Models

Planialtimetric data obtained by drone images (0.03 m) were processed by the Agisoft Metashape 1.6.2 software (AgisoftPhotoScan, 2018). The data obtained by drone images were integrated and assessed using the Global Mapper 22 Software [104]. The digital elevation model of the surfaces colonized by mangrove trees and herbs was acquired by photogrammetry according to drone images and verified by ground control points (GCP). The planialtimetric data of the GCPs were obtained by electronic theodolite and a Trimble Catalyst receiver, supported by a differential Global Navigation Satellite System with a decimeter accuracy (precision ± 10 cm). The Global Mapper 22 Software executed interpolation of the GCPs in areas of dense vegetation cover to record the tidal flat topography. Then, the Digital Terrain Model obtained for flats with dense vegetation was a product of the fusion of GCP interpolation from tidal flats with dense vegetation with the ground point gradients of non-vegetated flats extrapolated to flats below the dense vegetation cover. The vegetation height was obtained by the Combine/Compare Terrain Layers tool, which subtracts the Digital Surface Model from the Digital Terrain Model. A complete description of drone image processing may be found in [14,26,29,30].

### 4.4. Fieldwork and Sample Processing

Two sediment cores (NAT−3, S 5°40′18.33″/W 35°14′37.17″, 1 m deep, and NAT−5, S 5°40′17.40″/W 35° 14′35.70″, 1.5 m deep) were acquired via a Russian sampler in a transition restinga/mangrove (NAT−5, 2.5 m above mean sea-level), and in mangrove vegetation (NAT−3, 1.5 m above mean sea-level) (Figure 2a). Russian Peat sampler sidewall corers (5 cm × 50 cm) ensured no vertical compaction during sampling because they are a side-filling chambered-type sampler. This sampler enables one to drive the sampler to any point in the sediment profile in the closed (empty) position. When the target depth is attained, the handle is turned clockwise to sample as the pivotal cover plate supports the cutting action of the bore. As the sampler is turned 180 degrees, the sharpened edge of the bore longitudinally cuts a semi-cylindrical shaped sample until the opposite side of the cover plate is contacted. The sample can be recovered without risk of contamination by overlying sediments [105,106,107,108]. The sampling core sites were chosen according to the topography of the tidal flat, seeking a core that best represents the environmental history of the upper and lower limits of the tidal flat, but avoiding the proximity of channels that could have caused reworking of the sediments. Planialtimetric data of the sampling core sites were obtained using a GPS. The cores were cataloged, and images were registered in the field and kept in a cold room (4 °C).

### 4.5. Facies Description

The sediment cores were radiographed to characterize sedimentary aspects. Samples of ~0.5 g, obtained using a scale with a precision of 1 mg and at 10 cm intervals (46 samples) along the studied cores, were necessary to guarantee the quality of grain size analysis by laser diffraction (SHIMADZU SALD 2101) at the Laboratory of Coastal Dynamics of the Federal University of Pará UFPA (Brazil). The sampling considered the heterogeneity of the beddings to evidence the alternations in the grain size fractions, which was not always possible due to the millimeter thickness of the mud/sand layers. Hydrogen peroxide and ultrasound were applied to eliminate organic matter and disaggregate sediment particles, respectively. Grain sizes classification followed [109]: sand (2 mm–62.5 μm), silt (62.5–3.9 μm) and clay (3.9–0.12 μm). Sedimentary facies were described based on structures, lithology, color, and texture [110,111]. The sedimentary facies, pollen analysis, elementary, and isotopes data were assembled into facies associations to define a sedimentary environment [39]. Cluster analysis of pollen grains contributed to characterizing the facies associations.

### 4.6. Pollen Analysis

Forty-four sediment samples of 1 cm^3^ were selected every 5 cm along the cores (NAT−3 and NAT−5) for palynological analysis. The conventional analytical procedures of pollen in the samples followed the criteria of [112]. Pollen concentration was determined for each sample using an exotic *Lycopodium* spore capsule (grains/cm^3^). Pollen databases were used to help identify pollen and spores [113,114,115,116], and at least 300 pollen grains were counted for each sample. Pollen diagrams are exhibited as percentages. The taxa were classified into mangroves, herbs, tree/shrubs, and palms. Tilia software was used for the cluster analysis and pollen diagrams (Version 1.7.16) [117].

### 4.7. Isotopic Analysis

A total of 90 samples (10–60 mg) were obtained at 5 cm intervals along the cores. HCl (4%) was applied to remove carbonate [118]. The Stable Isotope Laboratory of the Center for Nuclear Energy in Agriculture (CENA/USP) provided the δ^13^C, δ ^15^N, total organic carbon (TOC), and total nitrogen (TN) analyses for the samples with an accuracy of 0.09% and 0.07%, respectively. Our δ^13^C and C/N data were analyzed according to sedimentary organic matter determined for C3 terrestrial plants (δ^13^C: −32–−21‰ and C/N > 12), C4 plants (δ^13^C: −17–−9‰ and C/N > 20) [60,62,119], freshwater algae (δ^13^C: −25–−30‰, and C/N: 6–10) [119,120], and marine algae (δ^13^C: −24–−18‰, and C/N: 6–10) [119]. Organic matter from terrestrial and aquatic plants have δ ^15^N values between ~0‰ and ~10‰, respectively [120]. Mangrove trees normally exhibit the following values: *Rhizophora* (δ^13^C: −31.8–−30.2‰), *Avicennia* (δ^13^C: −30.4–−29.1‰), and *Laguncularia* (δ^13^C: −30.4–−29.2‰) [38].

### 4.8. Radiocarbon Dating

In order to obtain a chronological model, three sediment samples (~10 g) were taken for C-14 dating. Predominantly muddy layers accumulated by vertical accretion were sampled for C-14 dating. Roots, plants debris, and seeds were physically removed from the sediment samples by a stereomicroscope to avoid carbon rejuvenation [121,122]. The sedimentary organic matter was chemically pretreated to remove carbonates and any more recent organic material, such as fulvic and/or humic acids, following the laboratory standard method defined in [118]. This process involved extracting residual material with 2% HCl at 60 °C for 4 h, washing with distilled water to neutralize the pH, and drying at 50 °C [122]. An accelerator mass spectrometer (AMS) at LACUFF (Fluminense Federal University) provided the C-14 dates. The radiocarbon dates were calibrated (cal yr BP) (2σ) using CALIB 8.2 and SHCal20 curve [123], and the text shows only the median of the calibrated ages (Table 1). In addition, the two radiocarbon dates along the core NAT−5 were input into RBacon to generate an age-depth model [124] using the SHCal20 curve [123], and the post-bomb SH Zone 1–2 curve [125] (Appendix A). This age-depth model uses Bayesian statistics to reconstruct accumulation histories for sedimentary deposits, allowing us to infer ages for stratigraphic levels that have not been dated [124], such as core NAT−5: ~2170 cal yr BP at 55 cm, and ~84 cal yr BP at 5 cm (Appendix A, Figure 4).

## 5. Conclusions

Multi-proxy data revealed mangrove migrations along a topographic profile of a tidal flat of the Ceará-Mirim estuary, northeastern Brazil, during the mid–late Holocene and Anthropocene. Mangroves expanded on the highest tidal flats with estuarine organic matter between ~4500 and 2870 cal yrs BP, under the influence of the mid-Holocene sea-level highstand. However, a mangrove contraction occurred after ~2870 cal yr BP with an increased contribution of C3 terrestrial plants, probably due to an RSL fall in the late Holocene. Mangroves have recolonized the highest flats of the study area during the last ~84 cal yr BP due to a relative sea-level rise, probably related to the end of the LIA, preceding the effects of anthropogenic emissions of CO_2_ in the atmosphere. However, significant mangrove areas were converted to fish farming before 1984 CE. Spatial-temporal analysis revealed a mangrove expansion since 1984 CE caused by mangrove recolonization of shrimp farming areas previously deforested for pisciculture, evidencing the resilience of these forests in the face of anthropogenic interventions.

## Figures and Tables

**Figure 1 plants-12-01721-f001:**
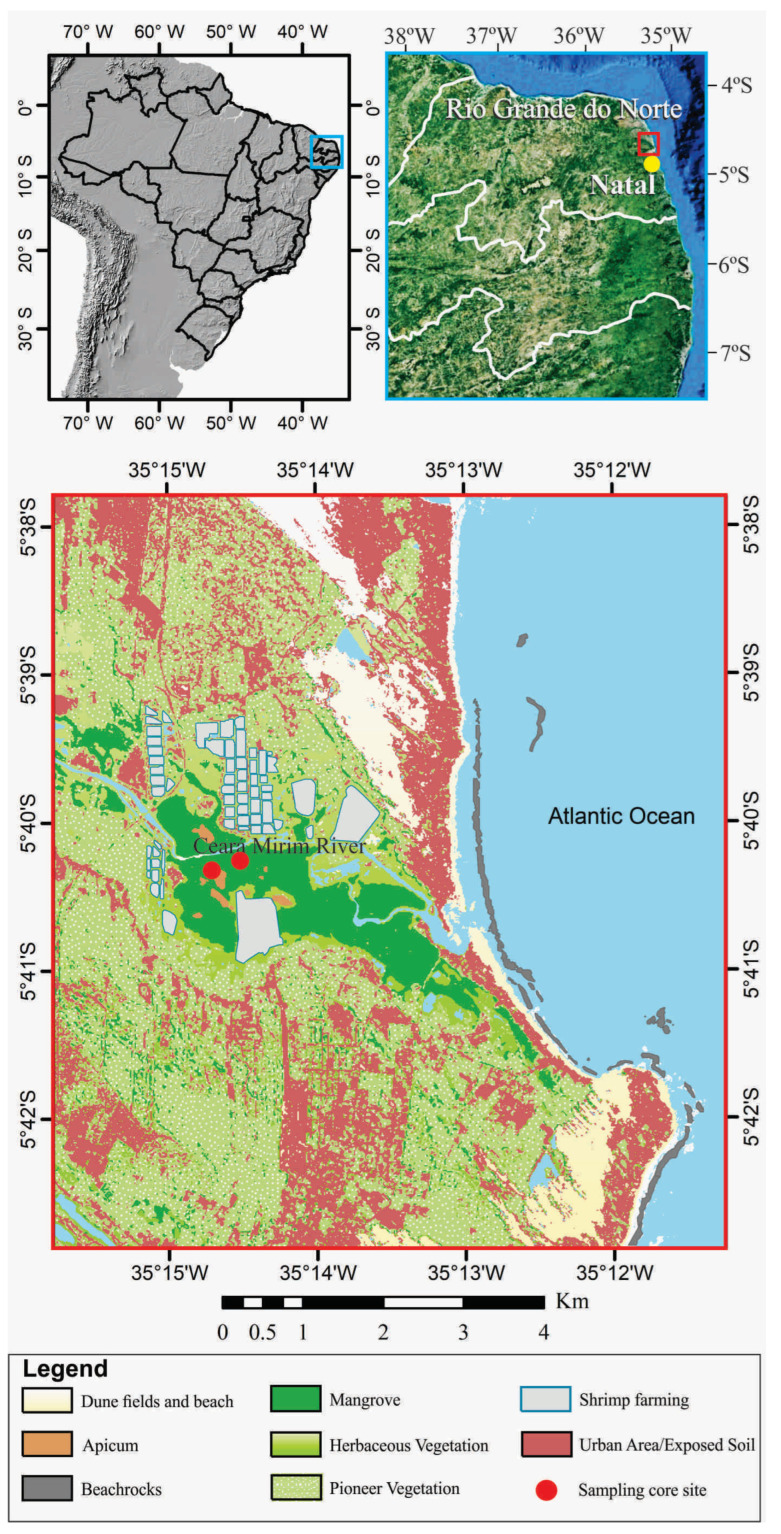
Location of the study area and vegetation units.

**Figure 2 plants-12-01721-f002:**
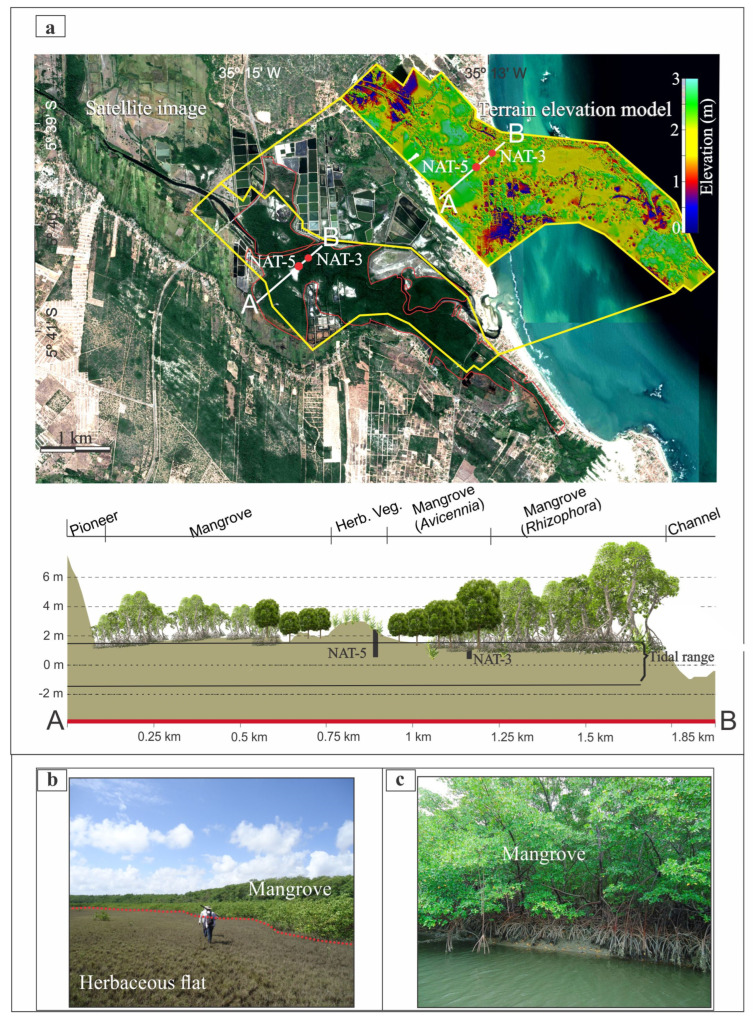
(**a**) Satellite image indicating the mangrove area, the digital terrain model, and the sampling core sites along a topographic profile (A–B) with vegetation units, (**b**) ground photo presenting the transition herbaceous flat and mangrove with *Avicennia* trees, (**c**) ground photo showing mangrove with *Rhizophora* trees along a channel.

**Figure 3 plants-12-01721-f003:**
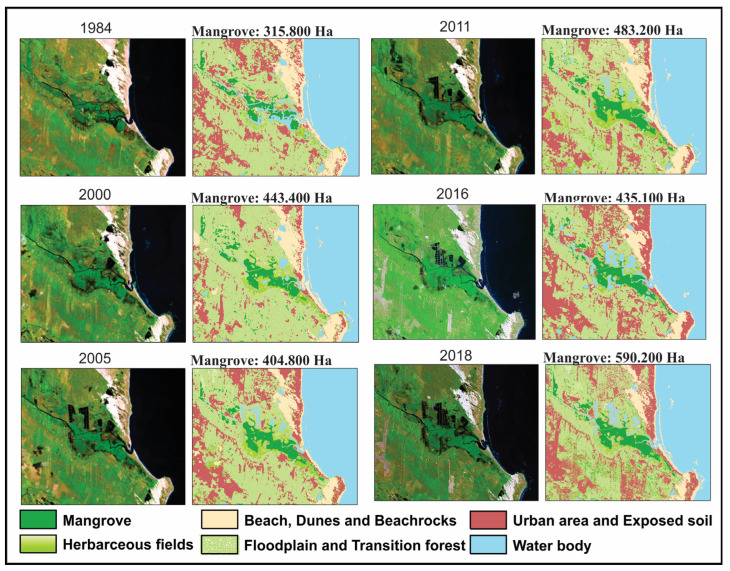
Multitemporal analysis between 1984 and 2018. Area calculations were performed for each class derived from the unsupervised classification.

**Figure 4 plants-12-01721-f004:**
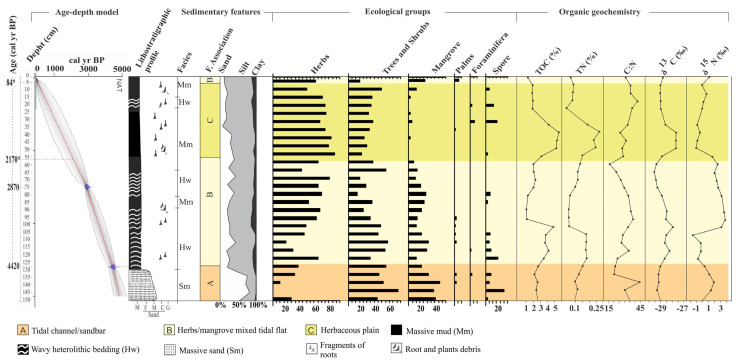
Summary results for core NAT−5, with variation depending on the depth of the core, showing chronological and lithological profiles with sedimentary facies, as well as ecological groups of pollen and geochemical variables. Pollen data are presented in the pollen diagrams as percentages of the total pollen sum.

**Figure 5 plants-12-01721-f005:**
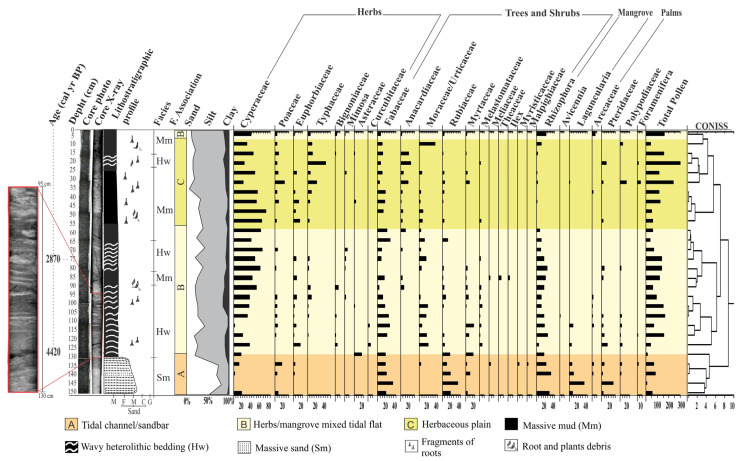
Pollen diagram of core NAT−5, with percentages of the most frequent pollen rates, age of the samples, and cluster analysis.

**Figure 6 plants-12-01721-f006:**
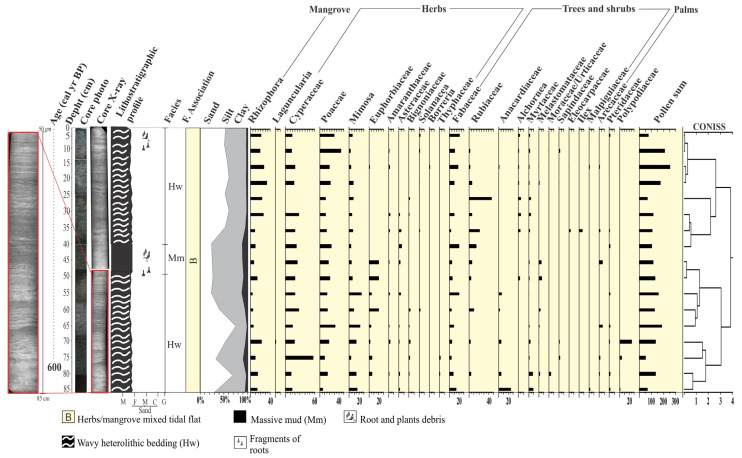
Pollen diagram of core NAT−3, with percentages of the most frequent pollen rates, age of the samples, and cluster analysis.

**Figure 7 plants-12-01721-f007:**
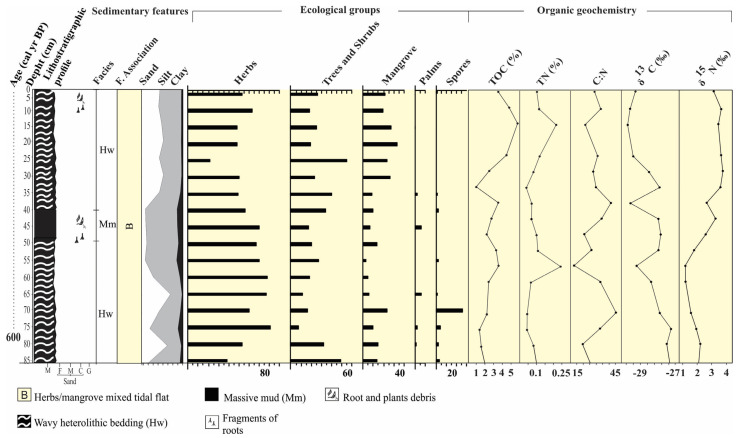
Summary results for core NAT−3, with variation depending on the depth of the cores, showing chronological and lithological profiles with sedimentary facies, as well as ecological groups of pollen and geochemical variables. Pollen data are presented in the pollen diagrams as percentages of the total pollen sum.

**Figure 8 plants-12-01721-f008:**
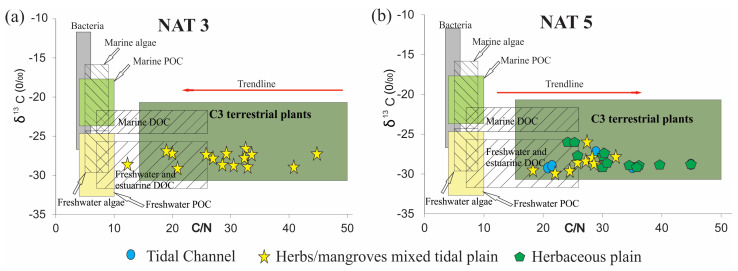
Diagram with the relationship between δ^13^C and C:N for the NAT−3 and NAT−5 cores according to the interpretations and data presented by [48,49] for the different sources of organic matter.

**Figure 9 plants-12-01721-f009:**
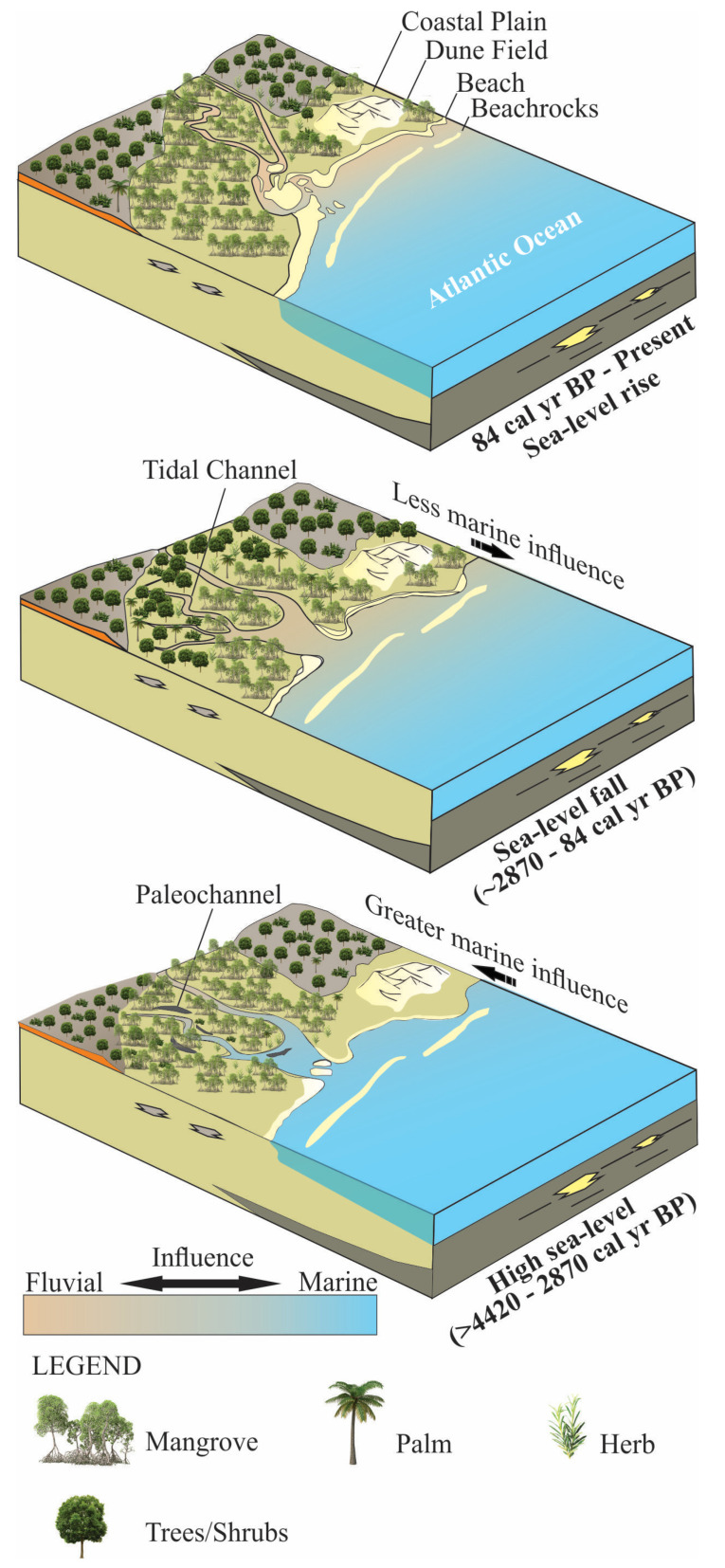
Eco–geomorphological models according to sea–level fall during the mid–late Holocene.

**Figure 10 plants-12-01721-f010:**
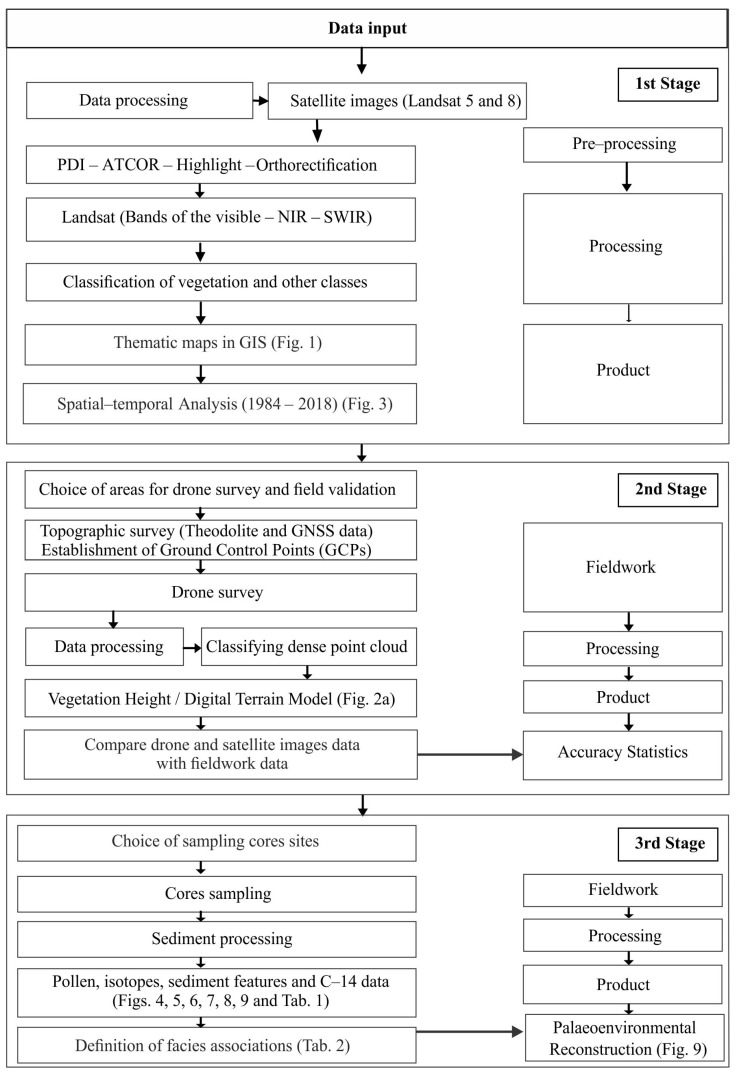
Methodology flow chart. Figure 1, Figure 2a, Figure 3, Figure 4, Figure 5, Figure 6, Figure 7, Figure 8 and Figure 9 and Table 1 and Table 2 are cited in this figure.

**Table 1 plants-12-01721-t001:** Sediment samples selected for the ^14^C dating with location, laboratory number, depth, material, and ages ^14^C years conventional BP, calibrated, and median.

Core	Cody Site and Laboratory Number	Depth (cm)	Material	Ages ^14^C Yr BP 1σ	Ages Cal Yr AD, 2σ	Median of Age Range (Cal Yr AD)
NAT 3	LACUFF-190615	70–80	Sed. organic matter	699 ± 35	557–615	600
NAT 5	LACUFF-190616	70–80	Sed. organic matter	2814 ± 29	2777–2962	2870
NAT 5	LACUFF-190617	125–135	Sed. organic matter	3997 ± 29	4291–4523	4420

**Table 2 plants-12-01721-t002:** Summary of the facies association with sedimentary characteristics, predominance of pollen groups, and geochemical data.

Facies Association	Facies Descripition	Pollen Predominance	Geochemical Data	Interpretation
A	Fine to medium sand (Sm).	Trees, shrubs, mangroves, and herbs	δ^13^C = −29.7 a −26.8‰ δ^15^N = −0.84 a 1.6‰ TOC = 3.74 a 17.58% TN = 0.12 a 0.54% C:N = 21.2 a 35.4	Tidal channel/sandbar
B	Massive mud (Mm) with greenish gray color and wave-type heterolithic bedding (Hw). Greenish gray color. Bioturbation with plant fragments and root marks.	Trees and shrubs, mangroves, herbs, and palm trees	δ^13^C = −29.2 a −28.7‰ δ^15^N = −0.91 a 2.91‰ TOC = 3.44 a 17.58% TN = 0.06 a 0.43% C:N = 27.6 a 34.14	Herbs/mangrove mixed tidal flat
C	Heterolithic lenticular bedding (Hl). Greenish gray color. Bioturbation with plant fragments and root marks.	Trees and shrubs, mangroves, herbs, and palm trees	δ^13^C = −29 a −28.87‰ δ^15^N = −1.39 a 2.55‰ TOC = 4.25 a 20.4% TN = 0.43 a 0.83% C:N = 24.5 a 28.67	Herbaceous plain

## Data Availability

Data will be made available upon request.

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
