# Peer review of "Assessment the Impacts of Sea-Level Changes on Mangroves of Ceará-Mirim Estuary, Northeastern Brazil, during the Holocene and Anthropocene"

_plants, 2023, doi:10.3390/plants12081721_

Round 1

Reviewer 1 Report

see attached pdf file

Author Response

Please find attached the revised version of the paper entitled “Assessment the impacts of sea level changes on mangroves of northeastern Brazil during the Holocene and Anthropocene”, by Nunes et al.,.

We appreciated all the comments and suggestions provided by the anonymous reviewers, which helped us to improve the earlier version of this manuscript. We carefully reviewed the manuscript to clarify all points raised. Parts of the text have been rewritten and new references/text have been added as suggested. They are all indicated in yellow color in the annotated version that we have attached. In the Response to Reviewers file, we respond to each comment individually.

Reviewer #1: Comments:

In this manuscript the development of a mangrove environment from the later mid Holocene to the present is investigated. The investigation area is the estuary of the Ceará - Mirim River at the Rio Grande do Norte coast north of the city Natal. The data base for this research are mainly Landsat 5 and satellite images and 2 sediment cores with a length of 1m respectively 1.5 m. The laboratory analyses which have been applied are standard methods. The main topic is mangrove migration as a response to sea level fluctuations and climate (precipitation / temperature) changes. Mangroves show their widest extent and migration upriver during sea level highstand fom 4500 – 2870 cal BP. The sea level drop after 2870 cal BP leads to a seaward migration of the mangrove environment which is shown by an increase of pollen percentage in the seaward core (NAT-3) since the last 600 years. An interesting aspect for the migration of mangroves seaward is the discussion about an increase in rainfall and fluvial discharge pushing the mangrove belt. However, some information about the seasonal fluctuation would be helpful to manifest this assumption. The strength of this manuscript is the area, which has never been investigated before. For this reason, a publication in plants can be considered. However, the biggest concern is the poor database with only 2 sediment cores and 3 C-14 dates.

Authors: We appreciate the analysis and comments. We have answered each comment individually below.

Reviewer #1: Structure of the manuscript: Normally the order of the different of the different chapters is: Introduction, regional settings, methods, results, discussion, conclusion. It is always good to read something about the methods which were applied before reading something about results. Results and discussion might be combined.

Authors: This is the structure of papers published by Journal Plants: 1. Introduction; 2. Results; 3. Discussion; 4. Materials and Methods; 5. Conclusions; Acknowledgments; and References. Therefore, the manuscript complies with the journal's standards.

Reviewer #1:  Objectives are poorly written. What is the authors Hypothesis?

Authors: According to Ribeiro et al. (2018), mangroves in the Ceará-Mirim River have an ecological history influenced mainly by autogenic mechanisms related to tidal channel dynamics, instead of allogenic processes related to climate changes since the Mid-Holocene. Our hypothesis is that the studied mangroves may have responded to changes in the relative sea level and river discharge during the Holocene. In addition, depending on the core sampling sites, the sign of these changes may be preserved through biogeochemical indicators in the stratigraphy (lines 54 – 65).

Reviewer #1:  Regional settings are poor. Size of the estuary, water and sediment discharge? How far does salinity and tidal range reach into the estuary? What’s about the sediment distribution? Is there a gradient from the tidal flats to the mangrove environment?

Authors: There are no studies on the flow and fractions of sediment grains size released by this river. However, Costa (2005) indicated the Ceará-Mirim river basin has 2,635.7 km² and flows in a west-east direction, with a length of 120 km crossing the coastal plateau/plain. It is influenced by semidiurnal saline and dynamic tides with a mean spring and neap tidal range of ~2.30 m and 0.85 m, respectively. Near the mouth, it presents a transition zone under marine and fluvial influence, dominated by sand-clay and silty sediments (Costa 2005; Pfaltzgraff 2010). The coastal plain is characterized by brackish water vegetation represented by mangroves on muddy tidal flats. Herbaceous vegetation occurs on highest sector of the tidal flats. Upstream, sandy silt and muddy sediments spread over the fluvial plain mainly dominated by “várzea” vegetation (swampland seasonally and permanently inundated by freshwater). The limit between várzea and mangrove vegetation depends on the estuarine salinity gradients (Lara and Cohen 2006; Ribeiro et al. 2018). (lines 451-461).

Reviewer #1:  Results. The comparison of satellite images shows an expansion of the mangrove area. Only 3,4 lines as results for this topic is really poor. Please describe a little bit more in which direction mangroves are expanding. There seems to be a fluctuation in the size of mangroves areas. Where (in which regions) are these changes? Always in the same areas or is it fluctuating over the whole areas.

Authors: The mangrove expansion was not progressive and constant mainly on higher limits of tidal flats (~1.7 – 2 m amsl), where Avicennia trees are leading this mangrove migration onto highest tidal flats. The transition zones show advances and retreats of the man-grove/herbaceous vegetation, where there is the coexistence of Poaceae, Cyperaceae and Avicennia shrubs. It is noteworthy that isolated and elevated sandy tidal flats (~2 m amsl), presenting a circular morphology with herbaceous vegetation and without human inter-ference, also show an intermittent advances and retreats of Avicennia shrub limits, but with an expansion trend between 1984 and 2018. By contrast, a progressive mangrove expansion has occurred mainly onto lower limits of tidal flats (~1 - ~1.7 m amsl), with abandoned shrimp farms. These lower surfaces have been consistently invaded mainly by Rhizophora. (lines 107 – 117).

Reviewer #1: What are the uncertainties in your calculation of the size of mangrove environments?

Authors: The objects smaller than the pixel size (900 m2, 0.09 ha) are indistinguishable. Therefore, the mapping of mangrove/herbaceous vegetation and mangrove/abandoned shrimp farms transition will be limited by the uncertainties caused by the pixels with mixed coverages (e.g., mangrove, herbs, soil, and water). The spectral response of these coverages depends on the average reflectance involving the different targets in the pixel (30 m) (Pop et al. 1994). (lines 485–490).

Reviewer #1: Radiocarbon dates and sedimentation rates: Sediment in mangrove environments are highly bioturbated by plants (the root system) and organisms. Some information is necessary for the sediment cores. A photo with high resolution should be shown in the manuscript so that the reader gets an impression how the core looks like. Additionally, you should show some of your radiographs.

Authors: Photos and radiography along the analyzed cores are presented in figures 5 and 6. The radiography of these cores allowed us to identify internal sedimentary structures. As recorded in Figures. 5 and 6, it would not be possible to evidence such structures without radiography. The heterolytic beddings show a continuous vertical sediment accretion with some bioturbation but no significant sedimentary reworking that weakens the paleoenvironmental reconstruction. Thus, we assume these sediments, pollen content and sedimentary organic matter originated from the moment of establishment of the depositional environment with terrestrial and aquatic vegetation. Regarding the palynology, we can consider two components—pollen from ‘‘local’’ vegetation, and background pollen from ‘‘regional’’ vegetation (Janssen 1966; Andersen 1967; Sugita 1994). The transition between the local and regional is gradual and depends on each pollen taxon (Davis 2000; Behling et al. 2001)The isotopic and elemental characteristics of sedimentary organic matter along the studied cores depend on the in situ vegetation and input of fluvial and marine organic matter (Dittmar et al. 2001, 2006; Matos et al. 2020). (lines 141-152).

Reviewer #1:  The data base for age control with only 3 samples from 2 cores is very poor and not convincing. How can you justify that there is no age conversion with only 1 C-14 datum in sediment core NAT-3?

Authors: Based on ~500 radiocarbon dates, sedimentation rates obtained along 200 cores (3 m depth) sampled from Brazilian, USA, Puerto Rico tidal flats have oscillated between 0.1 and 5 mm/yr (mean ~1 mm/yr) (e.g. Cohen et al. 2005b, a, 2008, 2009a, b, 2012, 2020; Lara and Cohen 2009; Guimarães et al. 2010, 2012; Smith et al. 2011; França et al. 2012, 2014; Franca et al. 2016; Moraes et al. 2017; Rodrigues et al. 2021, 2022; Yao et al. 2022). Regarding the studied cores, the estimated sedimentary rates are 1.1 mm/year (80-0 cm, NAT-3, Figures 6 and 7), 0.3 mm/year (130-75 cm, NAT-5) and 0.25 mm/year (75-0 cm, NAT-5, Figures 4 and 5). (lines 122 – 124). In addition, the RSL fall during the late-Hoocene favored a higher sedimentation rate for the core NAT-3 (1.1 mm/yr) than NAT-5 (0.25 -0.3 mm/yr) due to the difference in topography of the sampling core sites and mangrove structure. The core NAT-3 (~1.5 m amsl) was sampled from a tidal flat mainly dominated by Rhizophora, while NAT-5 (2.5 m amsl) site presented a flat with herbaceous vegetation and some Avicennia shrubs nearby. The sediment supply and accommodation space, which depends on the topography relative to the mean sea-level, control the sedimentation rates (Schlager 1993; Soreghan and Dickinson 1994). Besides, the mangrove structure and productivity also affect the sedimentation rates along a topographic gradient (Furukawa and Wolanski 1996; Kathiresan Kandasamy 2003; Ellis et al. 2004; McKee et al. 2007; Phillips et al. 2017; Matos et al. 2020). Species with prop roots, such as Rhizophora that dominates lower tidal flats, favor vertical accretion more than species with pneumatophores, such as Avicennia, which occurs on higher flats (Furukawa and Wolanski 1996; Krauss et al. 2003). Probably the RSL fall contributed to the Rhizophora mangrove establishment and proportionally higher sedimentation rates on the lower flats (lines 323-337). Therefore, considering the sampling core sites in topography and mangrove zonation, the sedimentation rates are within the expected range.

Reviewer #1:  Line 110 (references). References are given from NE-Brazil, mainly from the Amazon River environment. I recommend to have a look to an area between Natal and the Amazon River: Szcygielski et al., 2014. Evolution of the Parnaíba Delta (NE Brazil) during the late Holocene, DOI 10.1007/s00367- 014-0395-x.

Authors: Thank you very much, we added that reference (lines 288 – 210).

Reviewer #1:  Facies description: Some data about grain sizes are missing.

Authors: We added the percentages of the sediment grain size (lines 165-166, 184, 210).

Reviewer #1:  The data in the column for the sediment in figures 3 – 6 are poor.

Authors: We have added photographs and radiographs of the two studied cores and information about the sedimentary features that can be described (Figures. 5 and 6).

Reviewer #1: How was this considered in the sampling strategy?

Authors: The sampling core sites were chosen according to the topography of the tidal flat, seeking cores that best represent the environmental history of the upper and lower limits of the tidal flat, but avoiding the proximity of channels that could have caused reworking of the sediments (lines 521). Sediments were sampled at 10 cm (grains size) and 5 cm intervals (pollen, isotope, elemental analysis) (lines 545-552, 555–566). Predominantly muddy layers accumulated by vertical accretion were sampled for C-14 dating (lines 570–571).

Reviewer #1: Table 3, facies A. Interpretation is a tidal channel, but there are no convincing arguments. Are there any indicators for currents or other indicator for a channel environment?

Authors: Facies A presents massive sand (Sm) with a gradual transition to wavy heter-olithic beddings (Hw) in the Facies B, suggesting that flow energy decreased (Edwards et al. 2005; Miall 2006). These facies may occur in various depositional settings (Rossetti and Thales 2008; Rossetti et al. 2012), but considering the studied sedimentary environment with tidal flats, tidal/estuarine channels, and sandbars, under the influence of a meso-tide (2 – 4 m, Short (1991)), it would be reasonable to propose that transition was caused by a typical tidal channel dynamics building up an upward-fining succession, with thick sand deposition succession at the base, including subtidal channel-filling, topped by intertidal muddy deposits. Alternatively, subtidal sandbars may have been exposed to a decreased flow energy and gradually began to accumulate wavy heterolithic bedding. (lines 235 – 244, Table 3)

Reviewer #1:  Line 175: The occurrence of roots and plants in the sediment core are mentioned. Are the positions of the roots vertical or horizontal? If their position is vertical, how can this be interpreted? What does it mean for your age control?

Authors: Roots arranged horizontally and vertically partially obliterated the sedimentary structures, impairing the identification of heterolytic beddings on radiographs (lines 185–187). Roots, plants debris and seeds were physically removed from the sediment samples by a stereomicroscope to avoid carbon rejuvenation (Scharpenseel and Becker-Heidmann 1992; Pessenda et al. 2001). The sedimentary organic matter was chemically pretreated to remove carbonates and any more recent organic material such as fulvic and/or humic acids following the laboratory standard method as defined in (Pessenda et al. 2012). (lines 570 – 575).

Reviewer #1: How were the sediment cores taken? Is there compaction during the coring process?

Authors: Two sediment cores were acquired via a Russian sampler. Russian Peat sampler sidewall corers ensure no vertical compaction during sampling (Moore et al. 1991; USEPA 1999; Ellison and Strickland 2015). (lines 513 – 521).

Reviewer #1: It would be good to see a sea level curve from this area where own data are included.

Authors: Many RSL curves have been proposed for northeastern Brazil, especially the one by Bezerra et al. (2003) that indicated a relatively rapid sea-level rise in Rio Grande do Norte between 7100–5800 cal yr BP and about 5000 cal yr BP reached 2.5–4.0 m above present sea level. After the mid-Holocene sea-level highstand (5000 cal yr BP), RSL fell immediately and eventually rose again about 2100–1100 cal yr BP, resulting in a second coastal retreat in the Holocene. This curve also indicates that a lower than actual RSL could have occurred in the Rio Grande do Norte. Our cores did not indicate RSL oscillations between 2100 and 1100 cal yr BP. However, these cores also do not allow refuting the possibility of a RSL oscillation during the late Holocene, because they may not have recorded this rapid and temporally short RSL rise. Following the topographic and estuarine gradient, new cores along the estuary could contribute to studies about late Holocene RSL oscillations along the Brazilian coast (Suguio et al. 1980, 1985; Angulo et al. 2006). (lines 338–346). This topic has been the subject of much debate, and we would like to emphasize that this work's objective is not to clarify whether there were RSL oscillations in the late Holocene.

Reviewer #1:  Line 285: Replace “last” against “upper”.

Authors: Ok, thanks. (line 268)

Reviewer #1:  Line 291: A sea level rise has no influence on the “frequency” of inundation, but maybe on the time of inundation.

Authors: In that case we respectfully disagree. Many works have been published to explain the effect of sea level rise on tidal inundation frequency and porewater salinity along the topographic gradients of tidal flats (Cohen and Lara 2003; Lara and Cohen 2006; Cohen et al. 2018, 2020a, 2021). The sentence has been rewritten to better clarify this issue.

Reviewer #1:  Additionally, you use the term “probably”, so you have no hard arguments. You continue in line 292 with “This process contributed”, which is not correct, as you are not sure – it is only “probably”.

Authors: The interpretation and discussion are based on robust multi-proxies: pollen profiles, isotopes and sedimentary features. However, we must be humble enough to accept other interpretations because the basis for the interpretations of paleo sea-level, paleoclimate and paleoenvironment are always proxies. Environmental proxy indicators have the potential to provide evidence for large-scale climatic, sea-level and depositional system changes. However, the interpretation of a proxy record is complicated by the presence of “noise” in which environmental information is immersed, and a variety of possible distortions of the underlying environmental information (e.g. Bradley 1999, 2014; Ren 1999), then high-resolution of a set of proxies should be used to provide detailed information on environmental back in time (Houghton et al. 2001). We believe that paleoenvironmental reconstructions based on facies association (e.g., pollen, isotopes, and sedimentary features) along stratigraphic profiles, integrated into the characterization of current vegetation and geomorphological units of the study area, have the potential to attenuate such “noise” in the signals of the paleoenvironmental indicators, making the interpretations more reliable (lines 221–232).

Considering the meaning of the word “Probably”, with all due respect, according to the website “www.vocabulary.com” (https://www.vocabulary.com/dictionary/probably) Probably (adverb) with considerable certainty; without much doubt. Therefore, the word “Probably” is well used in the text, as it reflects exactly our thought. We have considerable certainty, without much doubt about our interpretation based on the multi-proxies.

Reviewer #1:  Line 298: Do you really mean that the last little ice age lasted six to seven centuries as a global signal? This is still a debate, please mention this at least and give some references.

Authors: This sentence has been rewritten and references have been added (lines 418 – 421).

Reviewer #1:  Line 305: Please explain what do you mean with “psychculture”.

Authors: That was a spelling error. It was replaced by pisciculture (line 426).

Reviewer #1:  Line 329/330: Please do not write “tidal flats occupied by mangroves”. Tidal flats are not occupied by plants, as you show in your picture 1c. It is the tidal environment which is partly occupied by mangroves.

Authors: This sentence has been rewritten (line 461).

Reviewer #1:  Chapter 4.4: Your sediment cores are the base for your studies. Please give some more information about your sampling procedure. “Russian sampler” is not known internationally. How do you drive the core into the sediment? What is the core diameter? Is there compaction during the coring procedure?

Authors: Russian Peat sampler sidewall corers (5 cm x 50 cm) ensure no vertical compaction during sampling because it is a side-filling chambered-type sampler. This sampler enables one to drive the sampler to any point in the sediment profile in the closed (empty) position. When the target depth is attained, the handle is turned clockwise to sample as the pivotal cover plate supports the cutting action of the bore. As the sampler is turned 180 degrees, the sharpened edge of the bore longitudinally cuts a semi-cylindrical shaped sample until the opposite side of the cover plate is contacted. The sample can be recovered without risk of contamination or compaction by overlying sediments (Bricker-Urso et al. 1989; Moore et al. 1991; USEPA 1999; Ellison and Strickland 2015) (lines 513 – 524).

Reviewer #1:   Chapter 4.5: You took radiographs. You should show some of them.

Authors:  The radiographs are shown along the cores (Figures. 5 and 7).

Reviewer #1:   How can you preciously sample 0.5 g? Was your sampling stratigraphy on the amount of material or did you sample certain layers? The latter should give you a better information if you have wavy or lenticular bedding.

Authors: Samples of ~0.5 g, obtained using a scale with a precision of 1 mg at 10 cm intervals (46 samples) along the studied cores, were necessary to guarantee the quality of grain size analysis by laser diffraction (SHIMADZU SALD 2101) at the Laboratory of Coastal Dynamics of the Federal University of Pará UFPA (Brazil). The sampling considered the heterogeneity of the beddings to evidence the alternations in the grain size fractions, which is not always possible due to the millimeter thickness of the mud/sand layers (lines 529 – 535).

Reviewer #1:   Chapter 4.8: You refer to reference 84 for the description of the standard method. Unfortunately, there is not very much written about the sampling treatment.

Authors: We added information about the pre-treatment that precedes the radiocarbon analyzes (lines 578–586).

Reviewer #1:   Some more comments Figures: Figures: Figure 1 is poor and needs improvement. 1b should be a little bit bigger. The profile A – B is too small. The city Natal is mentioned in the text but nowhere shown in a figure.

Authors: Figure 1 has been split in two (Fig. 1 and Fig. 2) to enlarge figure 1b and the profile A-B. We highlight the location of Natal in Figure. 1.

Reviewer #1:    Figure 8: This figure looks nice; however, the lower figure shows the development from 4420 – 2870, which is a period of a slight sea level drop. How can you explain a greater marine influence under those conditions? Is there tectonic influence or a decrease in sediment discharge? You did not explain in your section 3.2.

Authors: Indeed, studies have indicated a gradual RSL fall in southeastern, northeastern and northern Brazil since the mid-Holocene sea-level highstand (e.g., Angulo et al. 2006; Cohen et al. 2020a) or may have occurred oscillations (Martin et al. 2003; Bezerra et al. 2003; Suguio et al. 2013; Boski et al. 2015). Considering the study area, the effects of the RSL fall on coastal vegetation and sedimentary organic matter with a reduction in marine influence and seaward mangrove migration occurred only after ~2870 cal yr BP. The lower mid-Holocene high sea-level stand (~1 m amsl) along the north and northeast Brazilian coast (Boski et al. 2015; Cohen et al. 2021), when compared with the southeastern and southern Brazilian coast (3 – 5 m amsl, Angulo et al. (2006; 2016); Toniolo et al. (2020)), may have caused this delay to record this RSL fall in the studied cores. In addition, the climatic component through fluvial discharge may have interfered with the effects of marine influence upstream (see section see 3.4. Climatic effects). (line 351).

Reviewer #1: References: 63 is poor. What is it?

Authors: That citation has been corrected.

Reviewer #1 Title: This title does not really reflect what is written in the manuscript. Anthropogenic effects are not really shown just mentioned in the final part of the manuscript. This part should be erased from the title or better discussed.

Authors: The title has been changed to “Assessment the impacts of sea level changes on mangroves of northeastern Brazil during the Holocene and Anthropocene”.

Reviewer 2 Report

The manuscript presents an innovative topic with the incorporation of satellite data and spatial analysis. The data presented are important for understanding coastal risks to sea level rise. However, I consider that there are some points where the authors can improve the data presentation.

1. I suggest that the authors include a broad discussion of the climatic factors and processes that lead to sea level changes in the study area, as well as an explanation of how these influence their results.

2. The CONISS analysis is elaborated but it is not mentioned what it was elaborated for. It is necessary to explain why it is not useful to establish the facies, or why the pollen zones were not determined. what is the reason for its omission? please explain.

3. The manuscript states in section 4.8 Radiocarbon dating:"The ages are shown as calibrated years (cal yr BP)  (2σ) using CALIB 8.2 and SHCal20 curve [85]. The text shows only the median of the calibrated ages (Table 1). All radiocarbon dates were input into Bacon for R to generate an age-depth model (Table S1) [86]";  but it is not understood why you used two different programs, Bacon in R is enough to generate the ages, if the idea is to compare dates, this is not clearly mentioned, it is necessary to clarify this. You also do not explain clearly how many dates were used for the model, how many were discarded and if there is a possible Hiatus.

4. The use of remote sensing is relevant, but the integration of paleoecological information with the results of spatial analysis is somewhat weak, I recommend reinforcing the argumentation in the introduction and discussion. 

Author Response

Title: " Assessment the impacts of sea level changes on mangroves of northeastern Brazil during the Holocene and Anthropocene”.

Please find attached the revised version of the paper entitled “Assessment the impacts of sea level changes on mangroves of northeastern Brazil during the Holocene and Anthropocene”, by Nunes et al.,.

We appreciated all the comments and suggestions provided by the anonymous reviewers, which helped us to improve the earlier version of this manuscript. We carefully reviewed the manuscript to clarify all points raised. Parts of the text have been rewritten and new references/text have been added as suggested. They are all indicated in yellow color in the annotated version that we have attached. In the Response to Reviewers file, we respond to each comment individually.

Reviewer #2 The manuscript presents an innovative topic with the incorporation of satellite data and spatial analysis. The data presented are important for understanding coastal risks to sea level rise. However, I consider that there are some points where the authors can improve the data presentation.

Authors: We appreciate the analysis and comments. We have answered each comment individually below.

Reviewer #2 1. I suggest that the authors include a broad discussion of the climatic factors and processes that lead to sea level changes in the study area, as well as an explanation of how these influence their results.

Authors: This has been done (lines 221 – 232; 266 – 277, 278 – 349).

Reviewer #2 2. The CONISS analysis is elaborated but it is not mentioned what it was elaborated for. It is necessary to explain why it is not useful to establish the facies, or why the pollen zones were not determined. what is the reason for its omission? please explain.

Authors: Cluster analysis, based on pollen percentages, contributed to defining facies associations along the core NAT-5, while this analysis did not show significant pollen percentage changes that justify more than one facies association along the core NAT-3 (line 156-159).

Reviewer #2  3. The manuscript states in section 4.8 Radiocarbon dating:"The ages are shown as calibrated years (cal yr BP) (2σ) using CALIB 8.2 and SHCal20 curve [85]. The text shows only the median of the calibrated ages (Table 1).

Authors: Correct, the dates are presented along the text as the median of calibrated ages range (Table 1). That sentence has been rewritten (lines 578–582).

Reviewer #2 All radiocarbon dates were input into Bacon for R to generate an age-depth model (Table S1) [86]"; but it is not understood why you used two different programs, Bacon in R is enough to generate the ages, if the idea is to compare dates, this is not clearly mentioned, it is necessary to clarify this. 

Authors: The two radiocarbon dates along the core NAT-5 were input into RBacon to generate an age-depth model (Blaauw and Christen 2011) using the SHCal20 curve (Hogg et al. 2020), and the post-bomb SH Zone 1–2 curve (Hua et al. 2013) (Table S1). This age-depth model uses Bayesian statistics to reconstruct accumulation histories for sedimentary deposits, allowing to infer ages for stratigraphic levels that have not been dated (Blaauw and Christen 2011), such as: core NAT-5: ~2170 cal yr BP at 55 cm, and ~84 cal yr BP at 5 cm (Table 2, supplementary material, Figure 3). (Lines 580 – 586).

Reviewer #2 You also do not explain clearly how many dates were used for the model, how many were discarded and if there is a possible Hiatus.

Authors: No dating has been discarded (lines 570 – 577). Erosive surfaces are normal along stratigraphic profiles, causing Hiatus. Then, the need for a detailed stratigraphic description using radiographs that suggest along the studied cores a continuous vertical accretion with heterolithic bedding without evidence of erosive surfaces or sediment reworking (lines 141–159, 221 – 232, 235 – 244, Figures. 6 and 8).

Reviewer #2 4. The use of remote sensing is relevant, but the integration of paleoecological information with the results of spatial analysis is somewhat weak, I recommend reinforcing the argumentation in the introduction and discussion. 

Authors: Thanks for the suggestion. It has been done (lines 65 – 74, 107 – 117).

We hope to have addressed all comments accordingly. We look forward to your positive evaluation and acceptance of the revised manuscript.

Best regards,

Marcelo Cohen

Federal University of Pará

Rua Augusto Corrêa, 01

Cep: 66075-110, Bairro: Guamá

Belém-Pará

Tel: 91-3201-7478

Cel: 91-8031-1300

E-mail: mcohen80@hotmail.com

www.ufpa.br/cpgg

Round 2

Reviewer 1 Report

I am impressed by the update of the manuscript, it has improved a lot. Most of the explanation a very understandable and it is good to see them in the text.

However, the biggest concern in the first review was the poor data base regarding age control by 14-C analyses. There have been only 3 dating for 2 sediment cores. There are no more 14-C data shown in the revised version. With this poor data base, it is impossible to discuss any sedimentation rates especially in an environment which is dominated by bioturbation and sediment re-deposition. The explanation given by the authors in the response do not solve the problem of insufficient data. It really does not make sense to rely on sedimentation rates from other areas far away from the investigation site presented here. This topic should be completely erased (erase as well sedimentation rates in tab. 1 but leave the points in fig. 4); but the dating is not useless. In combination with sedimentological and facies-analyses these 14-C data can be used for stratigraphical purpose. As up to now no knowledge about the mangrove environment, which is presented here, exists, it makes sense to publish all the other parts of the manuscript. This will as well not intervene with the changed title.

Some more specific comments:

Title: The change of the title is appreciated but it would make sense to include the specific region “Ceará-Mirim estuary“, as just northeastern Brazil is too general.

Line 66: Please write in full words, respectively be more precise, what you did “…multi-proxies analysis (pollen, elementary, and isotopes data) along…. Is it pollen-analysis? Elementary analysis (please write the elements) isotope data analysis?

Line 73: ….more comprehensively to characterize….

Line 330 – 337. When writing about the function of the root system of mangroves to trap sediment it is recommenden to have a look to Oceanography, Vol. 30 and a special issue of Continental Shelf Research, Vol. 147, both published in 2017. Both issues deal in general with the situation of the Mekong Delta and some articles in particular with the functioning of mangroves.

Line 343/344: Please be precise with your attribute: “…. Holocene, because they may not have recorded this rapid and temporally short RSL rise…“. A core cannot record.

Figure 5: Core photo and x-ray. In the figure no details can be seen as this is definitely too small! It is not more than an overview. I recommend to present some cutouts with a much better resolution, so that the reader can follow your arguments. It would be nice to see in a much better resolution the most important internal structures and the sedimentary sequences. The resolution, as presented, cannot convince the reader.

References:

Correct Ref. 28.: It is a Geomorphology -article.

Ref. 62: Can be erased, as ref. 63 is the 3rd edition, which is normally an improvement of earlier editions.

Ref. 65. Please give a doi-number

Ref. Please add the journal. 1 doi is enough to find the reference.

Ref. 70: Journal is missing

Ref. 108: Full stop after name

Ref. 142: Please check duplication in wording

Ref. 144, 145, 147,149: What are these references, Journals? MSc- or PhD thesis? Please keep the citation standard.

Ref. 159: What is this? Not a reference.

Ref.: 19, 95, 112, 144, 145, 146, 147, 149: These references are in the local language. Please check if this is necessary for this international journal.

Author Response

Reviewer #2: Comments:

I am impressed by the update of the manuscript, it has improved a lot. Most of the explanation a very understandable and it is good to see them in the text.

Authors: We appreciate the analysis and comments. We have answered each comment individually below.

Reviewer #2: Comments:

However, the biggest concern in the first review was the poor data base regarding age control by 14-C analyses. There have been only 3 dating for 2 sediment cores. There are no more 14-C data shown in the revised version. With this poor data base, it is impossible to discuss any sedimentation rates especially in an environment which is dominated by bioturbation and sediment re-deposition. The explanation given by the authors in the response do not solve the problem of insufficient data. It really does not make sense to rely on sedimentation rates from other areas far away from the investigation site presented here. This topic should be completely erased (erase as well sedimentation rates in tab. 1 but leave the points in fig. 4); but the dating is not useless. In combination with sedimentological and facies-analyses these 14-C data can be used for stratigraphical purpose. As up to now no knowledge about the mangrove environment, which is presented here, exists, it makes sense to publish all the other parts of the manuscript. This will as well not intervene with the changed title.

This will as well not intervene with the changed title.

Authors: As suggested, sedimentation rates have been removed from the text, table 1 and figures 4, 5, 6 and 7. Discussions involving sedimentation rates have also been removed.

Reviewer #2: Some more specific comments:

Title: The change of the title is appreciated but it would make sense to include the specific region “Ceará-Mirim estuary“, as just northeastern Brazil is too general.

Authors: The title has been changed.

Reviewer #2: Line 66: Please write in full words, respectively be more precise, what you did “…multi-proxies analysis (pollen, elementary, and isotopes data) along…. Is it pollen-analysis? Elementary analysis (please write the elements) isotope data analysis?

Authors: this has been done (Lines 66-67).

Reviewer #2:  Line 73: ….more comprehensively to characterize….

Authors: This sentence has been rewritten (lines 73 - 75).

Reviewer #2: Line 330 – 337. When writing about the function of the root system of mangroves to trap sediment it is recommenden to have a look to Oceanography, Vol. 30 and a special issue of Continental Shelf Research, Vol. 147, both published in 2017. Both issues deal in general with the situation of the Mekong Delta and some articles in particular with the functioning of mangroves.

Authors: This paragraph involving sedimentation rates and mangrove root systems has been removed.

Reviewer #2:  Line 343/344: Please be precise with your attribute: “…. Holocene, because they may not have recorded this rapid and temporally short RSL rise…“. A core cannot record.

Authors: This sentence has been rewritten (lines 323-325).

Reviewer #2:  Figure 5: Core photo and x-ray. In the figure no details can be seen as this is definitely too small! It is not more than an overview. I recommend to present some cutouts with a much better resolution, so that the reader can follow your arguments. It would be nice to see in a much better resolution the most important internal structures and the sedimentary sequences. The resolution, as presented, cannot convince the reader.

Authors: Part of the core radiographs were enlarged to highlight the sedimentary structures (Figures 5 and 6).

References:

Correct Ref. 28.: It is a Geomorphology -article.

Ref. 62: Can be erased, as ref. 63 is the 3rd edition, which is normally an improvement of earlier editions.

Ref. 65. Please give a doi-number

Ref. Please add the journal. 1 doi is enough to find the reference.

Ref. 70: Journal is missing

Ref. 108: Full stop after name

Ref. 142: Please check duplication in wording

Ref. 144, 145, 147,149: What are these references, Journals? MSc- or PhD thesis? Please keep the citation standard.

Ref. 159: What is this? Not a reference.

Ref.: 19, 95, 112, 144, 145, 146, 147, 149: These references are in the local language. Please check if this is necessary for this international journal.

Authors: References have been revised and updated. We reduced the references from 177 to 129.

We hope to have addressed all comments accordingly. We look forward to your positive evaluation and acceptance of the revised manuscript.

Best regards,

Marcelo Cohen

Federal University of Pará

Rua Augusto Corrêa, 01

Cep: 66075-110, Bairro: Guamá

Belém-Pará

Tel: 91-3201-7478

Cel: 91-8031-1300

E-mail: mcohen80@hotmail.com

www.ufpa.br/cpgg